# Rovibrational Analysis of the *ν*_1_, *ν*_4_, *ν*_1_ + *ν*_4_ and *ν*_1_ − *ν*_4_ Bands of ^13^CF_4_

**DOI:** 10.3390/molecules30214267

**Published:** 2025-11-01

**Authors:** Ons Ben Fathallah, Romain Terrier, Laurent Manceron, Cyril Richard, Vincent Boudon

**Affiliations:** 1Laboratoire Interdisciplinaire Carnot de Bourgogne (ICB), UMR 6303, Université Bourgogne Europe, CNRS, F-21000 Dijon, France; 2Université Paris Cité and Université Paris Est Créteil, CNRS, LISA, F-75013 Paris, France

**Keywords:** ^13^CF_4_, high-resolution spectroscopy, tensorial formalism, line position analysis

## Abstract

We present a high-resolution infrared spectroscopic study of four vibrational bands of the ^13^CF_4_ isotopologue: the symmetric stretching fundamental $\nu_{1}$, the triply degenerate bending mode $\nu_{4}$, the combination band $\nu_{1}+\nu_{4}$, and the hot band $\nu_{1}-\nu_{4}$. A global analysis was performed using a tensorial formalism adapted to the $T_{d}$ symmetry group, allowing for consistent modelling of rotational structures. Reduced energy levels were extracted and fitted simultaneously for the four levels, yielding precise spectroscopic constants. The derived parameters enhance the spectroscopic characterization of ^13^CF_4_, a species of interest for isotopic studies and environmental monitoring. A total of 8992 transitions were assigned in the investigated spectral regions. The quality of fit is confirmed by a root mean square (RMS) deviation of about 0.0022 cm^−1^, highlighting the accuracy of the effective Hamiltonian model. These results provide a robust framework for future line list development and integration into spectroscopic databases such as TFMeCaSDa, HITRAN and GEISA.

## 1. Introduction

Tetrafuoromethane (CF_4_), also known as PFC-14, is a potent greenhouse gas of considerable interest in both atmospheric chemistry and environmental science. This interest stems from its extraordinary chemical inertness, exceptionally long atmospheric lifetime—estimated at around 50,000 years—and a very high global warming potential ($GWP_{100}$) of approximately 6500–7380 relative to CO_2_ [1,2]. As the most abundant perfluorocarbon present in the atmosphere, with background mole fractions now exceeding 80 parts per trillion (ppt), CF_4_ levels have increased steadily since the beginning of the industrial era. This growth is largely attributed to anthropogenic emissions, primarily from aluminium smelting via the Hall–Héroult process and from semiconductor manufacturing through plasma etching [3,4]. Given its extreme atmospheric persistence, CF_4_ remains in the atmosphere for geological timescales once released.

The most intense infrared-active band of CF_4_ arises from the asymmetric stretching vibrational mode ($\nu_{3}$), located near 1282 cm^−1^. This band has been the subject of extensive high-resolution spectroscopic investigations, leading to precise determinations of line positions and intensities [1,5,6]. As a result, it is well represented in major spectroscopic databases such as HITRAN [7]. Despite the extensive spectroscopic characterization of CF_4_, data concerning its isotopologue ^13^CF_4_ remain scarce. Only a limited number of studies have examined its rovibrational structure in detail. The first high-resolution measurements of the $\nu_{4}$ bending mode date back to the early 1980s, providing foundational insights into its vibrational behaviour [8]. Subsequent work focused on pressure broadening and line shape modelling, further refining our understanding of this fundamental band [9] but only for ^12^CF_4_. Despite numerous infrared studies of CF_4_, vibrational modes that are Raman-active have also attracted attention due to their sensitivity to molecular symmetry and anharmonic effects. In this context, Tabyaoui et al. [10] performed high-resolution stimulated Raman and CARS spectroscopy on the $\nu_{1}$ and $2\nu_{2}$ ($A_{1}$) bands of ^12^CF_4_. Their work yielded refined spectroscopic constants and revealed, for the first time, the high-resolution structure of the $2\nu_{2}$ overtone, with rotational transitions assigned up to J=40. These results significantly improved the description of Raman-allowed vibrational bands and contributed to a more accurate determination of fundamental molecular parameters. More recently, Martinez et al. carried out a combined Raman and infrared spectroscopic investigation, enabling rovibrational assignments for several vibrational modes of ^13^CF_4_ [11]. Recent progress in high-resolution techniques has significantly improved the spectroscopic description of CF_4_ and its isotopologues. Boudon et al. [6] revisited the $\nu_{3}$ region near 7.8 µm using high-resolution Fourier-transform infrared (FTIR) spectroscopy, achieving resolutions up to 0.0025 cm^−1^ under both room-temperature and jet-cooled conditions. Their analysis accounted for strong Coriolis coupling between the $\nu_{3}$ and $2\nu_{4}$ bands and extended to high rotational levels (J=56). The resulting synthetic linelist was included in the HITRAN 2008 and GEISA 2009 databases.

In a more recent study [12], the same authors used high-resolution stimulated Raman spectroscopy to investigate the $\nu_{1}$, $2\nu_{1}-\nu_{1}$, $\nu_{2}$, $2\nu_{2}$, and $3\nu_{2}-\nu_{2}$ bands of CF_4_ at 135 and 300 K. This work enabled an accurate determination of the C–F equilibrium bond length, $r_{e}=1.31588\left(6\right)$ Å. Finally, Simon et al. [13] employed terahertz cavity ring-down spectroscopy (THz-CRDS) to probe more than 50 pure rotational transitions of CF_4_ in the $\nu_{3}$ vibrational state with sub-MHz precision. Their high-resolution measurements enabled the full resolution of the tetrahedral splitting patterns, allowing for a precise determination of the dipole transition moment, ${\tilde{\mu }}_{3,3}=106.38\left(53\right)$ mD. A global Hamiltonian model was constructed to fit a comprehensive set of 25,863 rovibrational transitions, which have been incorporated into the TFMeCaSDa spectroscopic database.

However, a comprehensive high-resolution analysis of the $\nu_{1}$, $\nu_{4}$, $\nu_{1}+\nu_{4}$, and $\nu_{1}-\nu_{4}$ bands of the ^13^CF_4_ isotopologue is still lacking. In the present study, we provide a detailed high-resolution spectroscopic investigation of these four bands. The spectra are analysed using a tensorial formalism, allowing for a global fit of rovibrational transitions and the extraction of reduced energy levels. This approach reveals the vibrational structure and underlying symmetry effects of the molecule. The resulting spectroscopic parameters are expected to support future atmospheric modelling and contribute to major databases such as CaSDa, HITRAN and GEISA.

## 2. Results

### 2.1. Line Positions Analysis

The assignment of all spectral features reported in this work was carried out using SPVIEW software, 2.0.2 version, while the spectral modelling and line fitting were performed with the XTDS package [14,15,16,17], which incorporates the tetrahedral tensorial formalism within its STDS module. Due to the high density of transitions and unresolved overlapping transitions in the spectra, many theoretical transitions contribute to a single experimental line. Consequently, a single observed line often corresponds to a cluster of transitions that cannot be resolved experimentally. This explains why the number of assigned transitions considerably exceeds the number of observed spectral peaks. In order to reliably reproduce and fit the experimental spectra of ^13^CF_4_, we adopted a systematic approach: all terms of the effective Hamiltonian were expanded up to the sixth order. The ground state parameters were fixed to those previously derived for ^12^CF_4_ and used for building the TFMeCaSDa database [18] (we can notice that the TFMeCaSDa uses several sets of ground state parameters for the different polyads that are calculated; the differences between these sets are marginal, but this will require harmonization in a future update), assuming their limited sensitivity to isotopic substitution. The effective parameters for the $\nu_{4}$ and $\nu_{1}$ fundamental bands were fitted and exhibit only minor deviations compared to values from Refs. [6,10], as reported in Table 1 and Table 2, respectively. In contrast, the parameters associated with the other vibrational bands—namely $\nu_{1}+\nu_{4}$ and $\nu_{1}-\nu_{4}$—were determined here for the first time and are summarized in Table 2. A global fit involving all these bands was conducted to achieve a consistent and accurate spectroscopic model. In the following, we detail assignments and fit results of each for the four bands included in this global fit. A complete list of assignments for the four studied bands is provided in the Appendix A.

#### 2.1.1. $\nu_{4}$ Band

To analyse the line positions of the $\nu_{4}$ band of ^13^CF_4_, we used an infrared spectrum in the spectral range from 580 to 680 cm^−1^. This region is characterized by a very congested *Q*-branch. The $\nu_{4}$ band was considered in this work as an isolated band, and the weak overlapping bands present in this region were neglected.

Using the STDS software package Version 2023.04 from the XTDS framework, and after several iterations of least-squares fitting, we successfully assigned 2647 transitions up to J=75, by adjusting 27 effective Hamiltonian parameters. These parameters have good overall consistency with those reported in Ref. [6] for ^12^CF_4_ (apart, of course, from the band center which is isotopically shifted), as presented in Table 1.

Figure 1 provides an overview of the experimental and simulated spectra of the $\nu_{4}$ band, including a zoom-in on the *Q*-branch. The figure highlights the limited similarity in this congested region, where line density is particularly high. Figure 2 shows the residuals (observed minus simulated) for the fitted line positions, clearly indicating that deviations tend to increase with both wavenumber and rotational quantum number *J*.

Figure 3 presents a selected part of the *P*-branch, illustrating significantly better agreement between observed and calculated spectra. The lower panel displays the corresponding residuals, confirming the improved accuracy in this spectral region. Moreover, the upper panel of Figure 3 demonstrates a very good overall similarity between the experimental and simulated spectra.

#### 2.1.2. $\nu_{1}$ Band

To analyse the $\nu_{1}+\nu_{4}$ and $\nu_{1}-\nu_{4}$ bands, we constructed a polyad scheme that requires the inclusion of the $\nu_{1}$ band, as described at the end of Section 3.2. To fit the $\nu_{1}$ band together with the other bands, we used the same stimulated Raman scattering spectra as in Ref. [11], which were already presented in the introduction.

For this band, two spectra recorded at two different temperatures were used. For the spectrum recorded at 135 K, 729 transitions were assigned with rotational quantum numbers *J* ranging from 3 to 44. For the second spectrum, recorded at 300 K, 258 lines were assigned with *J* values from 45 to 80.

The fitting procedure for the $\nu_{1}$ band resulted in an average RMS deviation of $5.41\;\times \;10^{-4}$ cm^−1^. Table 2 presents the resulting parameters, compared to Ref. [11]. The differences with this reference are explained by the fact that, in this one, only the $v_{1}=1$ and $v_{1}=2$ levels were fitted, and the fixed (non-determinable) parameters were not the same). Figure 4 shows acn overview of both spectra along with the simulation.

#### 2.1.3. $\nu_{1}+\nu_{4}$ and $\nu_{1}-\nu_{4}$ Bands

##### $\nu_{1}-\nu_{4}$ Hot Band

The $\nu_{1}$–$\nu_{4}$ hot (or difference) band corresponds to a vibrational transition between the first excited level of the symmetric stretching mode ($\nu_{1}=1$) and the first excited level of the triply degenerate bending mode ($\nu_{4}=1$) of ^13^CF_4_. Within the $T_{d}$ point group, the $\nu_{1}$ mode exhibits $A_{1}$ symmetry, while the $\nu_{4}$ mode belongs to the $F_{2}$ irreducible representation.

Due to the relatively low thermal population of the $\nu_{1}=1$ level at room temperature, this hot band exhibits weak intensity yet remains clearly observable in the spectrum. A preliminary simulation was performed using an effective Hamiltonian based on the spectroscopic constants previously derived from the previous analyses of the $\nu_{1}$ and $\nu_{4}$ fundamental bands (see Table 1 and Table 2).

Figure 5 presents a comparison between the experimental spectrum and the simulated spectrum in the 230–320 cm^−1^ range. A good agreement is observed, with a root-mean-square (RMS) deviation on the order of $3\times 10^{-4}$ cm^−1^, and the residuals’ deviations are shown in Figure 6. The most intense lines visible in the lower panel correspond to contributions from residual H_2_O. This water spectrum was simulated under the same experimental conditions as those used for CF_4_ with data from HITRAN2020 [7].

A global fit including all three vibrational states (ν=0, $\nu_{1}=1$, and $\nu_{4}=1$) was then performed. A total of 2258 transitions were assigned and fitted, with rotational quantum numbers ranging from J=2 to J=68.

##### $\nu_{1}$ + $\nu_{4}$ Combination Band

The analysis of the $\nu_{1}+\nu_{4}$ combination band of ^13^CF_4_ was performed in the spectral region between 1480 and 1580 cm^−1^. This region presents a relatively complex structure due to the presence of several other weak vibrational bands. However, these bands exhibit much lower intensities and were therefore excluded from the present analysis. Their contribution to the overall line profile is negligible and does not affect the accuracy of the fitted parameters or the consistency of the global spectroscopic model. These unassigned lines appear in Figure 7 as weak experimental features that are absent from the simulation.

Despite the spectral congestion, about 3000 transitions were successfully assigned using the tensorial formalism implemented in the XTDS software 0.6.0. A total of 26 effective Hamiltonian parameters were adjusted to reproduce the observed line positions, with assignments extending up to rotational quantum numbers J=80. Figure 8 presents a comparison between the experimental and simulated spectra within this spectral range.

The quality of the fit highlights the reliability of the model used to describe this combination band. The fitted parameters are listed in Table 3, and the residuals between the observed and calculated line positions remain within an RMS of about 1.3 × 10^−3^ cm^−1^. These residuals are illustrated in Figure 9, which shows their distribution as a function of the wavenumber. To our knowledge, this is the first detailed modelling of the $\nu_{1}+\nu_{4}$ band of ^13^CF_4_ using a global effective Hamiltonian approach, contributing to an improved understanding of its high-resolution infrared spectrum.

Table 4 gives the fit statistics for the full global fit of line positions of the four bands under consideration in this work.

### 2.2. Energy Levels

In addition to the line position analysis, we derived the reduced effective rovibrational energy levels for each vibrational state investigated. The reduced wavenumber definition is:(1)$${\tilde{\nu }}_{\mathrm{red}}=\tilde{\nu }-\sum_{\Omega }t_{\left\{GS\}\{GS\right\}}^{\Omega (0,0A_{1})}\;{\left(J(J+1)\right)}^{\Omega /2}=\tilde{\nu }-B_{0}J\left(J+1\right)+D_{0}J^{2}{\left(J+1\right)}^{2}-\ldots$$
where $B_{0}$, $D_{0}$, etc., are ground state values for the parameters related to scalar effective Hamiltonian terms.

Figure 10 illustrates a presentation of energy upper levels for each vibrational band—$\nu_{4}$, $\nu_{1}$, $\nu_{1}+\nu_{4}$, and $\nu_{1}-\nu_{4}$—the calculated reduced energy levels exhibit excellent sampling of the experimentally observed levels accessed via assigned transitions. This consistency reflects the reliability of the fitted parameters and supports the validity of the tensorial formalism used. The resulting energy landscape provides a robust foundation for future theoretical and experimental investigations of ^13^CF_4_.

## 3. Experimental and Theoretical Methodology

### 3.1. Experimental Section

Several absorption spectra of ^13^C-labelled carbon tetrafluoride (^13^CF_4_) were recorded using a Bruker IFS 125 HR high-resolution Fourier transform spectrometer (FTS) with different optical cells at the AILES beamline of the SOLEIL synchrotron [19,20] (Bruker, Billerica, MA, USA). The setup included a 418 mm focal length collimator, the synchrotron radiation or a globar source, different beamsplitters and detectors (see Table 5). The ^13^CF_4_ gas sample was provided by Prof. D. Bermejo (Sigma Aldrich, St. Louis, MO, USA, 95 % purity) and used without further purification [12].

The varying parameters for each spectrum are detailed in Table 5. The resolutions mentioned in the table refer to those in the Bruker software 3.4, defined as 0.9/MOPD. Interferograms are collected without apodization (Boxcar option) and post-treated with a zero-filling factor of 8. Transmittance spectra were obtained by dividing the sample spectra by a zero-absorption spectrum recorded with the evacuated cells at 0.06 cm^−1^ resolution. Frequency axis calibration was performed using CO_2_, H_2_O impurity lines, and a high-resolution N_2_O spectrum with identical instrument parameters was used to complete the calibration for the $\nu_{4}$ region up to 630 cm^−1^. For the FIR region, we used 32 H_2_O lines between 475 and 116 cm^−1^, an additional set of 92 N_2_O lines and, in the mid-infrared range, 10 CO_2_ lines around 2300 cm^−1^ and 61 H_2_O lines between 1800 and 1375 cm^−1^. Line positions were compared to those from the HITRAN database [7], achieving an RMS deviation of about $3\times 10^{-4}$ cm^−1^ after calibration.

### 3.2. Theoretical Model

Carbon tetrafluoride (CF_4_), a tetrahedral spherical top molecule, exhibits four fundamental vibrational modes: one non-degenerate mode of $A_{1}$ symmetry ($\nu_{1}$), a doubly degenerate mode of *E* symmetry ($\nu_{2}$), and two triply degenerate modes of $F_{2}$ symmetry ($\nu_{3}$ and $\nu_{4}$).

For the spectroscopic analysis, we employ the tensorial formalism developed in Dijon for spherical top molecules. This methodology is based on the effective Hamiltonian approach, as described in [21], and allows for a systematic expansion of both the Hamiltonian and the transition moment operators. These expansions can be carried out to any desired order and are applicable to specific vibrational levels or entire polyads, facilitating the computation of line positions and intensities.

In the case of an XY_4_ molecule, vibrational levels are usually organized into a series of polyads, denoted $P_{k}$ (k=0,…,n), where $P_{0}$ corresponds to the ground state. After appropriate contact transformations, the effective Hamiltonian can be expressed in the following general form:(2)$$H=H_{\left\{P_{0}\equiv \mathrm{GS}\right\}}+H_{\left\{P_{1}\right\}}+\cdots +H_{\left\{P_{k}\right\}}+\cdots +H_{\left\{P_{n-1}\right\}}+H_{\left\{P_{n}\right\}}.$$
Terms such as $H_{\left\{P_{k}\right\}}$ contain rovibrational operators whose matrix elements vanish within the basis set of polyads $P_{k^{\prime }}$ with $k^{\prime }<k$. The effective Hamiltonian for a given polyad $P_{n}$ is obtained by projecting the total Hamiltonian H onto the Hilbert subspace associated with $P_{n}$; that is:(3)$$H^{\langle P_{n}\rangle }=P^{\langle P_{n}\rangle }HP^{\langle P_{n}\rangle }=H_{\left\{\mathrm{GS}\right\}}^{\langle P_{n}\rangle }+H_{\left\{P_{1}\right\}}^{\langle P_{n}\rangle }+\cdots +H_{\left\{P_{k}\right\}}^{\langle P_{n}\rangle }+\cdots +H_{\left\{P_{n-1}\right\}}^{\langle P_{n}\rangle }+H_{\left\{P_{n}\right\}}^{\langle P_{n}\rangle }.$$

The different terms are written in the form:(4)H{Pk}=∑allindexest{s}{s′}Ω(K,n,Γ)βVε{s}{s′}Ωv(ΓvΓv′)Γ⊗RΩ(K,n,Γ)(A1).

In this equation, $t_{\left\{s\right\}\left\{s^{\prime }\right\}}^{\Omega \left(K,n,\Gamma \right)\Gamma_{v}\Gamma_{v}^{\prime }}$ are the parameters to be determined, whereas Vε{s}{s′}Ωv(ΓvΓv′)Γ and $R^{\Omega (K,n,\Gamma )}$ are vibrational and rotational operators of respective degrees $\Omega_{v}$ and Ω. Their construction is described in Ref. [21]. Once again, the vibrational operators have non-zero matrix elements only within the $P_{k}^{\prime }\leq k$ basis sets. The factor β is introduced to ensure that scalar terms with symmetry $\Gamma =A_{1}$ are consistent with conventional expressions such as $B_{0}J^{2}$, among others. The order of each term in the expansion is defined by $\Omega +\Omega_{v}-2$.

A detailed correspondence between the Hamiltonian parameters used in this formalism and the more conventional notations found in the literature is provided in [22]. This correspondence is also explicitly illustrated for the vibrational bands analysed in this work, as presented in Table 1 and Table 2 (see the last section).

This Hamiltonian expansion framework allows for a systematic treatment of any polyad system. In order to analyse the various infrared bands studied here, we define a restricted polyad scheme for which $P_{0}$ is the vibrational ground state (GS), $P_{1}$ is the $v_{4}=1$ (so-called “$\nu_{4}$”) level, $P_{2}$ is the $v_{1}=1$ (“$\nu_{1}$”) level and $P_{3}$ is the $v_{1}=v_{4}=1$ (“$\nu_{1}+\nu_{4}$”) level. Thus, the following effective Hamiltonians are required:The GS effective Hamiltonian:(5)$$H^{\langle \mathrm{GS}\rangle }=H_{\left\{\mathrm{GS}\right\}}^{\langle \mathrm{GS}\rangle }.$$The $\nu_{4}$ effective Hamiltonian:(6)$$H^{\langle \nu_{4}\rangle }=H_{\left\{\mathrm{GS}\right\}}^{\langle \nu_{4}\rangle }+H_{\left\{\nu_{4}\right\}}^{\langle \nu_{4}\rangle }.$$The $\nu_{1}$ effective Hamiltonian:(7)$$H^{\langle \nu_{1}\rangle }=H_{\left\{\mathrm{GS}\right\}}^{\langle \nu_{1}\rangle }+H_{\left\{\nu_{4}\right\}}^{\langle \nu_{1}\rangle }+H_{\left\{\nu_{1}\right\}}^{\langle \nu_{1}\rangle }.$$The $v_{1}+v_{4}$ effective Hamiltonian:(8)$$H^{\langle \nu_{1}+\nu_{4}\rangle }=H_{\left\{\mathrm{GS}\right\}}^{\langle \nu_{1}+\nu_{4}\rangle }+H_{\left\{\nu_{4}\right\}}^{\langle \nu_{1}+\nu_{4}\rangle }+H_{\left\{\nu_{1}\right\}}^{\langle \nu_{1}+\nu_{4}\rangle }+H_{\left\{\nu_{1}+\nu_{4}\right\}}^{\langle \nu_{1}+\nu_{4}\rangle }.$$

The expression of the rovibrational basis set and the construction of the effective dipole moment and polarisability operators used for intensity calculations are described elsewhere [21]. Since we do not study here absolute line intensities, we simply mention that these operators are developed here to the minimum possible order.

## 4. Conclusions

In this work, we performed a detailed spectroscopic study of the $\nu_{1}$, $\nu_{4}$, $\nu_{1}+\nu_{4}$, and $\nu_{1}-\nu_{4}$ bands of ^13^CF_4_, based on FTIR and Raman spectra recorded at room and low temperature. The $\nu_{1}$ and $\nu_{4}$ fundamental bands, as well as the $\nu_{1}+\nu_{4}$ and $\nu_{1}-\nu_{4}$ combination and difference bands, were analysed by determining line positions and vibrational band assignments. Using the XTDS and STDS software packages, we developed a global effective Hamiltonian model capable of accurately reproducing the observed transitions in several vibrational bands, including fundamental, combination and hot bands. The modelling enabled the assignment of a large number of experimental lines and the refinement of spectroscopic parameters for both vibrational and rotational levels.

The results significantly enhance the spectroscopic characterization of ^13^CF_4_, providing reliable line position data suitable for inclusion in spectroscopic databases such as HITRAN and GEISA. Future work will focus on the extension of the analysis to higher-energy bands and the incorporation of line intensities and pressure-broadening parameters, in order to build a complete and accurate spectroscopic dataset for remote sensing and atmospheric applications. Such data will also contribute to refining the CF_4_ potential energy surface obtained through quantum chemistry calculations.

## Figures and Tables

**Figure 1 molecules-30-04267-f001:**
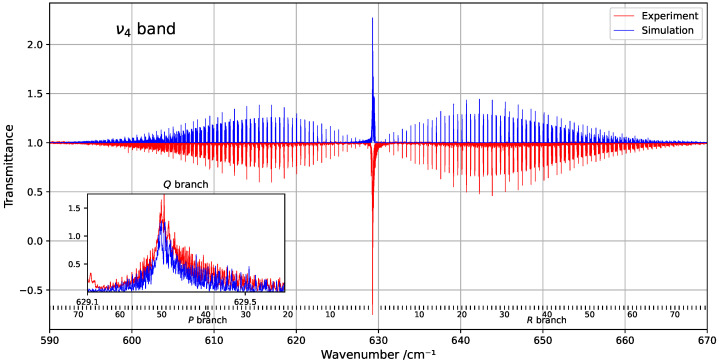
Overview of the $\nu_{4}$ band at 294 K, compared with the corresponding simulation. The rotational quantum numbers *J* are indicated at the bottom of the figure. A zoomed-in view of the congested *Q*-branch is shown in the inset on the left.

**Figure 2 molecules-30-04267-f002:**
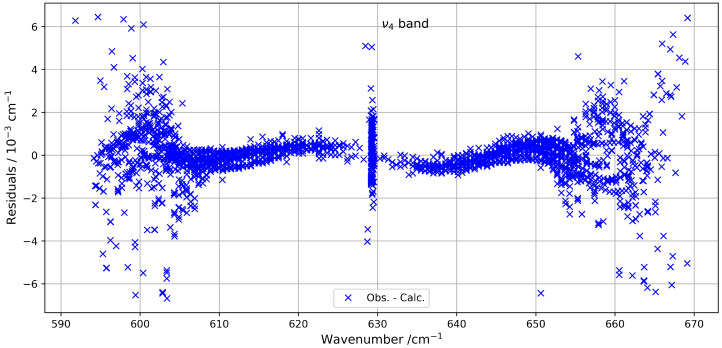
Line position fit residuals for the $\nu_{4}$ band.

**Figure 3 molecules-30-04267-f003:**
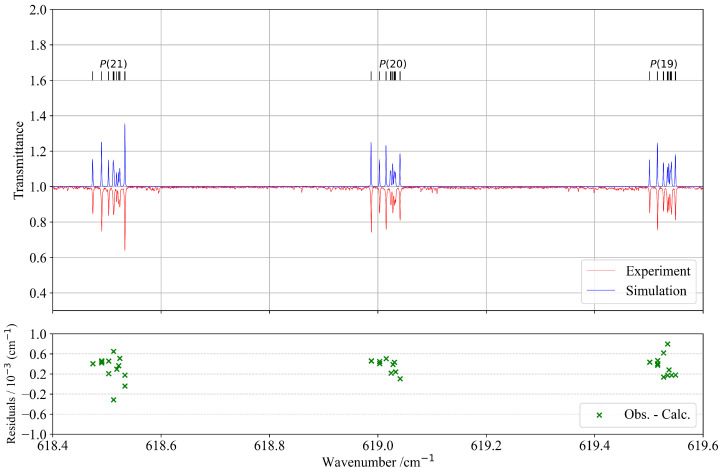
Detail in the *P*-branch of the $\nu_{4}$ band compared with the simulation. *J* quantum numbers with corresponding transitions are indicated at the top of the figure as ticks. The lower panel corresponds to residuals between the experimental and calculated positions of the assigned lines.

**Figure 4 molecules-30-04267-f004:**
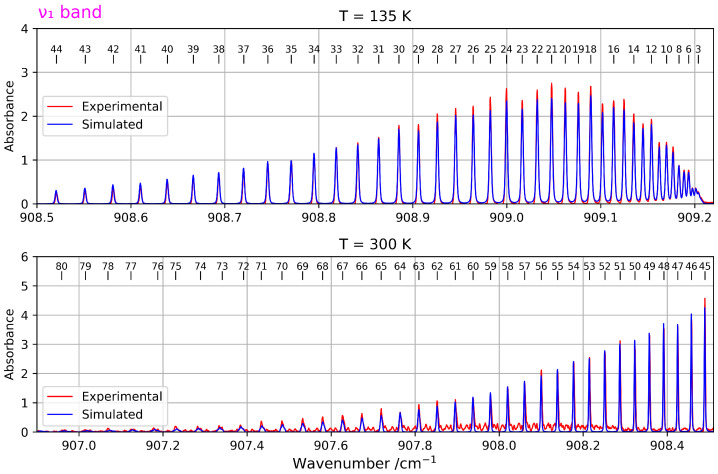
Overview of a zoomed-in view of the $\nu_{1}$ band (stimulated Raman spectrum) at two different temperatures The upper panel corresponds to the spectrum recorded at *T* = 135 K, while the lower panel shows the one at *T* = 300 K and provides access to higher *J* values.

**Figure 5 molecules-30-04267-f005:**
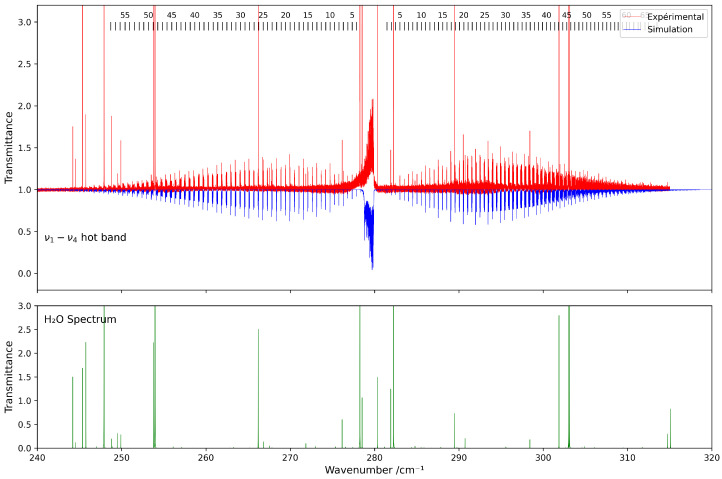
Comparison between experimental and simulated spectra of ^13^CF_4_ in the 230–320 cm^−1^ region ($\nu_{1}-\nu_{4}$ band). The lower panel shows the simulated spectrum of H_2_O under the same conditions.

**Figure 6 molecules-30-04267-f006:**
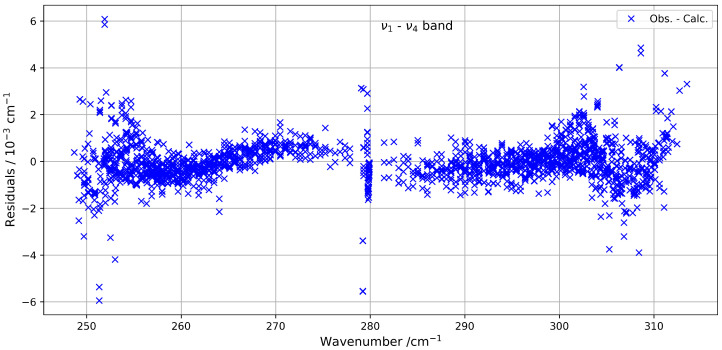
Residuals between the experimental and calculated positions of the $\nu_{1}-\nu_{4}$ hot band of ^13^CF_4_ in the 230–320 cm^−1^ spectral region.

**Figure 7 molecules-30-04267-f007:**
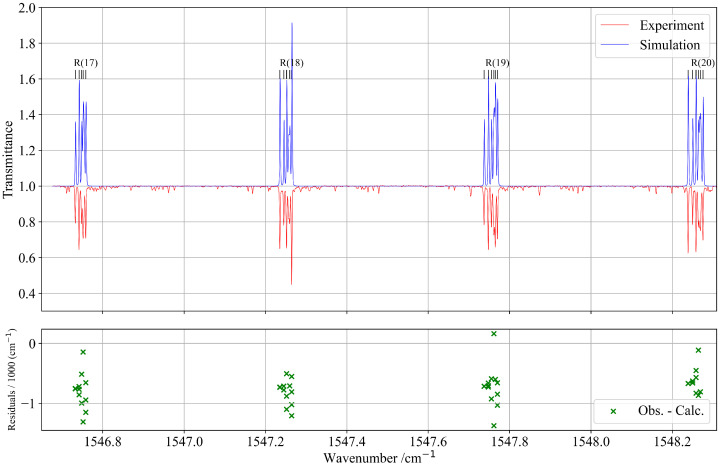
Zoomed-in view of a portion of the experimental and simulated spectra of the *R*-branch of the $\nu_{1}+\nu_{4}$ band. The lower panel shows the residuals of the assigned line positions in this region.

**Figure 8 molecules-30-04267-f008:**
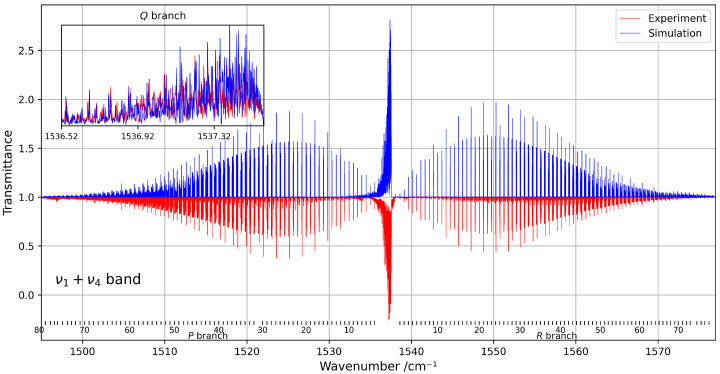
Overview of the $\nu_{1}+\nu_{4}$ band at 294 K, compared with the corresponding simulation. The rotational quantum numbers *J* are indicated at the bottom of the figure. A zoomed-in view of the congested *Q*-branch is shown in the inset on the left.

**Figure 9 molecules-30-04267-f009:**
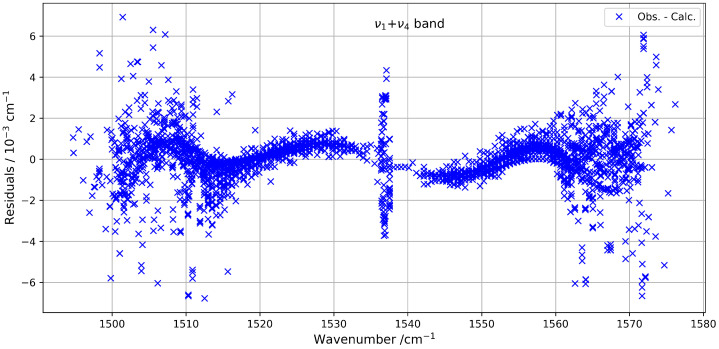
Residuals between observed and calculated line positions for the $\nu_{1}+\nu_{4}$ band.

**Figure 10 molecules-30-04267-f010:**
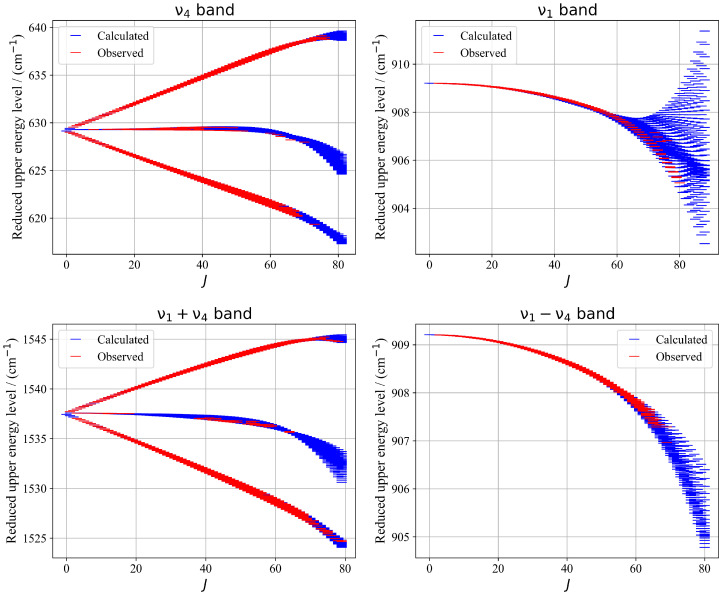
Zoomed-in view of a portion of the experimental and calculated spectra in the R-branch region of the $\nu_{1}+\nu_{4}$ band. The bottom panel shows the position residuals of the assigned lines in this region.

**Table 1 molecules-30-04267-t001:** Effective Hamiltonian parameter values for ground state (GS) and $v_{4}=1$ levels of ^13^CF_4_ and comparison with ^12^CF_4_ data from Ref. [6]. Parameters with no standard deviation in parentheses are fixed (see text). Notations are those of Equation (Equation 4).

Level	Order	Ω(K,nΓ)	{s} Γ	{$s^{\prime }$} $\Gamma^{\prime }$	This Work ^13^CF_4_	Ref. [6] ^12^CF_4_
					Value/cm^−1^	Value/cm^−1^
GS	0	$2(0,0A_{1})$	$0000A_{1}$	$0000A_{1}$	$1.9119296291\times 10^{-1}$	$1.9119312\left(46\right)\times 10^{-1}$
GS	2	$4(0,0A_{1})$	$0000A_{1}$	$0000A_{1}$	$-6.0892609629\times 10^{-8}$	$-6.13\left(13\right)\times 10^{-8}$
GS	2	$4(4,0A_{1})$	$0000A_{1}$	$0000A_{1}$	$-3.0206293797\times 10^{-9}$	$-3.028\left(25\right)\times 10^{-9}$
GS	4	$6(0,0A_{1})$	$0000A_{1}$	$0000A_{1}$	$8.8255632470\times 10^{-12}$	$9.07\left(59\right)\times 10^{-12}$
GS	4	$6(4,0A_{1})$	$0000A_{1}$	$0000A_{1}$	$-4.5496134947\times 10^{-13}$	$-4.572\left(63\right)\times 10^{-13}$
GS	4	$6(6,0A_{1})$	$0000A_{1}$	$0000A_{1}$	$-1.2086108035\times 10^{-13}$	$-1.205\left(21\right)\times 10^{-13}$
GS	6	$8(0,0A_{1})$	$0000A_{1}$	$0000A_{1}$	$-9.3653678951\times 10^{-16}$	$-9.61\left(87\right)\times 10^{-16}$
GS	6	$8(4,0A_{1})$	$0000A_{1}$	$0000A_{1}$	$3.3854359393\times 10^{-21}$	$-1.5\left(3.9\right)\times 10^{-19}$
GS	6	$8(6,0A_{1})$	$0000A_{1}$	$0000A_{1}$	$5.0876340920\times 10^{-19}$	$5.4\left(2.7\right)\times 10^{-19}$
GS	6	$8(8,0A_{1})$	$0000A_{1}$	$0000A_{1}$	$1.3833675655\times 10^{-18}$	$1.407\left(66\right)\times 10^{-18}$
$v_{4}=1$	0	$0(0,0A_{1})$	$0001F_{2}$	$0001F_{2}$	629.151605(55)	631.059247(87)
$v_{4}=1$	1	$1(1,0F_{1})$	$0001F_{2}$	$0001F_{2}$	$-2.832702\left(39\right)\times 10^{-1}$	$-2.948885\left(87\right)\times 10^{-1}$
$v_{4}=1$	2	$2(0,0A_{1})$	$0001F_{2}$	$0001F_{2}$	$1.0088\left(14\right)\times 10^{-4}$	$1.0118\left(29\right)\times 10^{-4}$
$v_{4}=1$	2	2(2,E)	$0001F_{2}$	$0001F_{2}$	$-8.803\left(19\right)\times 10^{-5}$	$-8.067\left(41\right)\times 10^{-5}$
$v_{4}=1$	2	$2(2,0F_{2})$	$0001F_{2}$	$0001F_{2}$	$1.8211\left(25\right)\times 10^{-4}$	$1.8099\left(63\right)\times 10^{-4}$
$v_{4}=1$	3	$3(1,0F_{1})$	$0001F_{2}$	$0001F_{2}$	$-4.531\left(16\right)\times 10^{-7}$	$-4.078\left(66\right)\times 10^{-7}$
$v_{4}=1$	3	$3(3,0F_{1})$	$0001F_{2}$	$0001F_{2}$	$-3.350\left(12\right)\times 10^{-7}$	$-5.478\left(32\right)\times 10^{-7}$
$v_{4}=1$	4	$4(0,0A_{1})$	$0001F_{2}$	$0001F_{2}$	$7.852\left(98\right)\times 10^{-9}$	$-1.20\left(18\right)\times 10^{-9}$
$v_{4}=1$	4	4(2,0E)	$0001F_{2}$	$0001F_{2}$	$1.680\left(28\right)\times 10^{-8}$	$-3.173\left(38\right)\times 10^{-8}$
$v_{4}=1$	4	$4(2,0F_{2})$	$0001F_{2}$	$0001F_{2}$	$-2.70\left(28\right)\times 10^{-9}$	$2.828\left(37\right)\times 10^{-8}$
$v_{4}=1$	4	$4(4,0A_{1})$	$0001F_{2}$	$0001F_{2}$	$2.991\left(14\right)\times 10^{-9}$	$2.121\left(49\right)\times 10^{-9}$
$v_{4}=1$	4	4(4,0E)	$0001F_{2}$	$0001F_{2}$	$-9.45\left(42\right)\times 10^{-9}$	$5.321\left(54\right)\times 10^{-8}$
$v_{4}=1$	4	$4(4,0F_{2})$	$0001F_{2}$	$0001F_{2}$	$-5.58\left(30\right)\times 10^{-9}$	$3.765\left(45\right)\times 10^{-8}$
$v_{4}=1$	5	$5(1,0F_{1})$	$0001F_{2}$	$0001F_{2}$	$1.340\left(30\right)\times 10^{-11}$	$1.34\left(13\right)\times 10^{-11}$
$v_{4}=1$	5	$5(3,0F_{1})$	$0001F_{2}$	$0001F_{2}$	$1.012\left(12\right)\times 10^{-10}$	$1.854\left(34\right)\times 10^{-10}$
$v_{4}=1$	5	$5(5,0F_{1})$	$0001F_{2}$	$0001F_{2}$	$1.243\left(14\right)\times 10^{-10}$	$2.447\left(40\right)\times 10^{-10}$
$v_{4}=1$	5	$5(5,1F_{1})$	$0001F_{2}$	$0001F_{2}$	$3.254\left(51\right)\times 10^{-11}$	$2.664\left(86\right)\times 10^{-11}$
$v_{4}=1$	6	$6(0,0A_{1})$	$0001F_{2}$	$0001F_{2}$	$-4.513\left(18\right)\times 10^{-12}$	$1.14\left(35\right)\times 10^{-13}$
$v_{4}=1$	6	6(2,0E)	$0001F_{2}$	$0001F_{2}$	$2.545\left(16\right)\times 10^{-12}$	$-2.160\left(38\right)\times 10^{-12}$
$v_{4}=1$	6	$6(2,0F_{2})$	$0001F_{2}$	$0001F_{2}$	$1.014\left(15\right)\times 10^{-12}$	$1.968\left(46\right)\times 10^{-12}$
$v_{4}=1$	6	$6(4,0A_{1})$	$0001F_{2}$	$0001F_{2}$	$8.998\left(19\right)\times 10^{-13}$	$8.10\left(39\right)\times 10^{-14}$
$v_{4}=1$	6	6(4,0E)	$0001F_{2}$	$0001F_{2}$	$8.48\left(23\right)\times 10^{-13}$	$3.108\left(61\right)\times 10^{-12}$
$v_{4}=1$	6	$6(4,0F_{2})$	$0001F_{2}$	$0001F_{2}$	$-8.51\left(24\right)\times 10^{-13}$	$2.259\left(84\right)\times 10^{-12}$
$v_{4}=1$	6	$6(6,0A_{1})$	$0001F_{2}$	$0001F_{2}$	$1.145\left(12\right)\times 10^{-13}$	$1.24\left(53\right)\times 10^{-14}$
$v_{4}=1$	6	6(6,0E)	$0001F_{2}$	$0001F_{2}$	$2.482\left(70\right)\times 10^{-13}$	$4.60\left(25\right)\times 10^{-13}$
$v_{4}=1$	6	$6(6,0F_{2})$	$0001F_{2}$	$0001F_{2}$	$-2.035\left(17\right)\times 10^{-12}$	$2.81\left(73\right)\times 10^{-13}$
$v_{4}=1$	6	$6(6,1F_{2})$	$0001F_{2}$	$0001F_{2}$	$6.64\left(42\right)\times 10^{-16}$	$6.49\left(36\right)\times 10^{-13}$

**Table 2 molecules-30-04267-t002:** Effective Hamiltonian parameter values for the $v_{1}=1$ level of ^13^CF_4_.

Level	Order	Ω(K,nC)	{s} C_1_	{$s^{\prime }$} C_2_	This Work	Ref. [11]
					Value/cm^−1^	Value/cm^−1^
$v_{1}=1$	0	$0(0,0A_{1})$	$1000A_{1}$	$1000A_{1}$	909.2081139(87)	909.20879(11)
$v_{1}=1$	2	$2(0,0A_{1})$	$1000A_{1}$	$1000A_{1}$	$-3.472835\left(87\right)\times 10^{-4}$	$-3.47657\left(11\right)\times 10^{-4}$
$v_{1}=1$	4	$4(0,0A_{1})$	$1000A_{1}$	$1000A_{1}$	$8.88\left(16\right)\times 10^{-11}$	$2.571\left(83\right)\times 10^{-10}$
$v_{1}=1$	4	$4(4,0A_{1})$	$1000A_{1}$	$1000A_{1}$	$-1.24\left(10\right)\times 10^{-11}$	$-1.53\left(13\right)\times 10^{-11}$
$v_{1}=1$	6	$6(0,0A_{1})$	$1000A_{1}$	$1000A_{1}$	0.00	$-1.829\left(93\right)\times 10^{-14}$
$v_{1}=1$	6	$6(4,0A_{1})$	$1000A_{1}$	$1000A_{1}$	$-3.774\left(82\right)\times 10^{-15}$	$-4.48\left(12\right)\times 10^{-15}$
$v_{1}=1$	6	$6(6,0A_{1})$	$1000A_{1}$	$1000A_{1}$	$-1.536\left(34\right)\times 10^{-15}$	$-1.95\left(42\right)\times 10^{-16}$

**Table 3 molecules-30-04267-t003:** Effective Hamiltonian parameter values for $v_{1}=v_{4}=1$ level of ^13^CF_4_.

Level	Order	Ω(K,nC)	{s} C_1_	{$s^{\prime }$} C_2_	Value/cm^−1^
$v_{1}=v_{4}=1$	0	$0(0,0A_{1})$	$1001F_{2}$	$1001F_{2}$	$-9.13537\left(92\right)\times 10^{-1}$
$v_{1}=v_{4}=1$	1	$1(1,0F_{1})$	$1001F_{2}$	$1001F_{2}$	$4.1612\left(64\right)\times 10^{-3}$
$v_{1}=v_{4}=1$	2	$2(0,0A_{1})$	$1001F_{2}$	$1001F_{2}$	$-8\left(3\right)\times 10^{-7}$
$v_{1}=v_{4}=1$	2	2(2,E)	$1001F_{2}$	$1001F_{2}$	$2.30\left(45\right)\times 10^{-6}$
$v_{1}=v_{4}=1$	2	$2(2,0F_{2})$	$1001F_{2}$	$1001F_{2}$	$-5.85\left(64\right)\times 10^{-6}$
$v_{1}=v_{4}=1$	3	$3(1,0F_{1})$	$1001F_{2}$	$1001F_{2}$	$-2.58\left(25\right)\times 10^{-8}$
$v_{1}=v_{4}=1$	3	$3(3,0F_{1})$	$1001F_{2}$	$1001F_{2}$	$3.39\left(16\right)\times 10^{-8}$
$v_{1}=v_{4}=1$	4	$4(0,0A_{1})$	$1001F_{2}$	$1001F_{2}$	$1.0\left(2\right)\times 10^{-9}$
$v_{1}=v_{4}=1$	4	4(2,0E)	$1001F_{2}$	$1001F_{2}$	$4.91\left(35\right)\times 10^{-9}$
$v_{1}=v_{4}=1$	4	$4(2,0F_{2})$	$1001F_{2}$	$1001F_{2}$	$-4.18\left(40\right)\times 10^{-9}$
$v_{1}=v_{4}=1$	4	$4(4,0A_{1})$	$1001F_{2}$	$1001F_{2}$	$2.426\left(34\right)\times 10^{-9}$
$v_{1}=v_{4}=1$	4	4(4,0E)	$1001F_{2}$	$1001F_{2}$	$1.60\left(51\right)\times 10^{-9}$
$v_{1}=v_{4}=1$	4	$4(4,0F_{2})$	$1001F_{2}$	$1001F_{2}$	$6\left(3\right)\times 10^{-10}$
$v_{1}=v_{4}=1$	5	$5(1,0F_{1})$	$1001F_{2}$	$1001F_{2}$	$4\left(3\right)\times 10^{-13}$
$v_{1}=v_{4}=1$	5	$5(3,0F_{1})$	$1001F_{2}$	$1001F_{2}$	$-1.79\left(16\right)\times 10^{-11}$
$v_{1}=v_{4}=1$	5	$5(5,0F_{1})$	$1001F_{2}$	$1001F_{2}$	$-2.33\left(17\right)\times 10^{-11}$
$v_{1}=v_{4}=1$	5	$5(5,1F_{1})$	$1001F_{2}$	$1001F_{2}$	$-1.33\left(70\right)\times 10^{-12}$
$v_{1}=v_{4}=1$	6	$6(0,0A_{1})$	$1001F_{2}$	$1001F_{2}$	$7.91\left(36\right)\times 10^{-13}$
$v_{1}=v_{4}=1$	6	6(2,0E)	$1001F_{2}$	$1001F_{2}$	$-2.43\left(23\right)\times 10^{-13}$
$v_{1}=v_{4}=1$	6	$6(2,0F_{2})$	$1001F_{2}$	$1001F_{2}$	$-4.01\left(26\right)\times 10^{-13}$
$v_{1}=v_{4}=1$	6	$6(4,0A_{1})$	$1001F_{2}$	$1001F_{2}$	$9.78\left(43\right)\times 10^{-14}$
$v_{1}=v_{4}=1$	6	6(4,0E)	$1001F_{2}$	$1001F_{2}$	$2.50\left(27\right)\times 10^{-13}$
$v_{1}=v_{4}=1$	6	$6(4,0F_{2})$	$1001F_{2}$	$1001F_{2}$	$8.50\left(38\right)\times 10^{-13}$
$v_{1}=v_{4}=1$	6	$6(6,0A_{1})$	$1001F_{2}$	$1001F_{2}$	$-4.17\left(15\right)\times 10^{-14}$
$v_{1}=v_{4}=1$	6	6(6,0E)	$1001F_{2}$	$1001F_{2}$	$2.56\left(67\right)\times 10^{-14}$
$v_{1}=v_{4}=1$	6	$6(6,0F_{2})$	$1001F_{2}$	$1001F_{2}$	$7.91\left(32\right)\times 10^{-13}$
$v_{1}=v_{4}=1$	6	$6(6,1F_{2})$	$1001F_{2}$	$1001F_{2}$	0.0

**Table 4 molecules-30-04267-t004:** Fit statistics for the various vibrational levels.

Level	Number of Assigned Transitions	$J_{\max}$	$d_{\mathrm{RMS}}/10^{-3}\;{\mathrm{cm}}^{-1}$
$\nu_{1}=1$	987	80	0.54
$\nu_{4}=1$	2783	75	1.27
$\nu_{1}+\nu_{4}=1$	2964	80	1.27
$\nu_{1}-\nu_{4}=1$	2258	68	0.88

**Table 5 molecules-30-04267-t005:** Experimental conditions for the various studied regions of ^13^CF_4_.

Vibrational Bands	Pressure (hPa)	Path Length (m)	Resolution (cm^−1^)	Beamsplitter	Source	Detector	Lowpass Filter (kHz)	Optical Velocity (cm/s)
$\nu_{1}+\nu_{4}$	0.515	0.845	0.00102	KBr/Ge	Globar + iris 1.15 mm	HgCdTe 4K + Cold filter ^1^	80	5.08
$\nu_{4}$	6.31	0.051	0.00102	KBr/Ge	Globar + iris 1.15 mm	HgCdTe 4K + Cold filter ^1^	80	5.08
$\nu_{1}-\nu_{4}$	20.05	96	0.0017	Mylar/Si	Synchrotron	Bolometer 4K + 700 cm^−1^ cold filter	40	2.53

^1^ Cold filter reference: [C]

## Data Availability

The original contributions presented in this study are included in the article/Appendix A. Further inquiries can be directed to the corresponding author.

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
