# Peer review of "Rovibrational Analysis of the ν1, ν4, ν1 + ν4 and ν1ν4 Bands of 13CF4"

_molecules, 2025, doi:10.3390/molecules30214267_

Round 1

Reviewer 1 Report

Comments and Suggestions for Authors

The paper entitled "Rovibrational analysis of the ν1, ν4, ν1 + ν4 and ν1 − ν4 bands of
13CF4" focuses on the analysis of four vibrational bands: the symmetric stretching fundamental ν1, the bending mode ν4, the combination band ν1 + ν4, and the hot band ν1 − ν4 of the 13CF4 isotopologue. A global analysis was carried out using a tensorial formalism for the Td symmetry group. Nearly 9000 transitions were assigned with an RMS of about 0.0022 cm−1

The research objective is presented in a clear way. The layout of the article is also clearly organized, consisting of an introduction, an experimental section, a theoretical model and a description of the analysis, ending up with conclusions and references. The work is nicely written.

Nevertheless, I have some remarks and comments. 

1) In Table 2, in which effective Hamiltonian parameter values for ground state (GS) and v4 =1 are compared between 12CF4 and 13CF4 , I noticed one inconcistency. The error of one of the parameters is higher than its value: −1.5(3.9)×10−19. It seems that this value was taken from the reference no 6, but anyway, it is not correct.

2) While describing a strategy for the line assignment and the fitting procedure, the authors write (line 150 and 151):

"The ground state parameters were fixed to those previously derived for 12CF4 and used for building the TFMeCaSDa database, assuming their limited sensitivity to isotopic substitution.

Was it enough to use the parameters for 12C or were they somehow refined after some cycles of fit?

3) The authors claim that (line 163):

"The ν4 band was considered in this work as an isolated band, and the weak overlapping bands present in this region were neglected."

On what basis were the weak overlapping bands omitted? Is it known what type of weak bands these are? Was any perturabtion analysis performed?

4) I found some minor typos in the text:

  • in line 150: "the effective Hamiltonian were expanded up to sixth order"; I think it should be up to the sixth order.
  • in Figure 5: a subscript is missing in ν1 − ν4.

In summary, the results of this work are extremly significant to the high-resolution spectroscopy community and would be even more valuable if line intensities were included in the analysis, but still worth publishing.

Reviewer 2 Report

Comments and Suggestions for Authors

This manuscript presents a thorough high-resolution spectroscopic investigation of four vibrational bands of the ¹³CFâ‚„ isotopologue. Using Fourier-transform infrared spectroscopy at the SOLEIL synchrotron, the authors performed a global rovibrational analysis with approximately 9000 transitions assigned, yielding refined spectroscopic constants with excellent accuracy. The study provides a valuable contribution to the spectroscopic characterization of ¹³CFâ‚„, complementing prior work on ¹²CFâ‚„ and supporting the inclusion of accurate parameters into major databases. The manuscript is technically sound, well organized, and clearly written.

I recommend publication of this manuscript after addressing the points below:

  1. Please ensure that all tables and figures are clearly labeled and described. For instance, the title of Figure 4 “Overview of a zoom of the ν1 band (stimulated Raman spectrum) at two different temperatures”could include the temperature explicitly, like “(Top: 135 K; bottom: 300K)” for ease of reference. Also, in Table 1, the first column “Spectral region” should refer to a spectral range in inverse centimeters, while the band labeling current shown should be labeled as “bands”.
  2. For calibration against water lines, please provide more details, such as the number and spectral range of water lines used for calibration, also, please cross-reference the version (if the same version is used) of HITRAN to Ref.[7] on page 4, line 95.
  3. Some of the parameters of the nu1=nu4=1 level in Table 4 have substantial uncertainties, as large as about half of the value itself, while for other levels, those constants are much better determined. Why? Typically, large uncertainties would suggest the assigned lines are not sensitive to the corresponding parameters, and the effective Hamiltonian should be modified. It’s worth to comment on that.
  4. There appear to be mixed uses of British and American English throughout the manuscript (e.g., behavior/behaviour, analyzed/analysed, modeling/modelling). Please ensure consistency in spelling style across the entire text.
  5. Please make sure that sub/superscribes are properly used for 12CF4/C4v/2nu2 etc. in the entire reference section.
